TECHNICAL RELEASE

# Sepsis-3 criteria in AmsterdamUMCdb: open-source code implementation

Tom Edinburgh[1,*], Stephen J. Eglen[1], Patrick Thoral[2], Paul Elbers[2] and Ari Ercole[3]

1  Department of Applied Mathematics and Theoretical Physics, University of Cambridge, Cambridge, UK
2  Department of Intensive Care Medicine, Amsterdam UMC, Amsterdam, The Netherlands
3  Cambridge Centre for Artificial Intelligence in Medicine and Division of Anaesthesia, Department of Medicine, University of Cambridge, Cambridge, UK

## ABSTRACT

Sepsis is a major healthcare problem with substantial mortality and a common reason for admission to the intensive care unit (ICU). For this reason, the management of sepsis is an important area of ICU research. A number of large-scale, freely-accessible ICU databases are available for observational research and the robust identification of septic patients in such data sets is crucial for research purposes, particularly for comparative studies between critical care sub-populations which may vary around the world. However, data structures are poorly standardised due to inevitable variances in clinical electronic health record system vendor and implementation as well as research database design choices. Robust and well-documented cohort selection (such as patients with sepsis) is crucial for reproducible research. In this work, we operationalise the Sepsis-3 definition on the AmsterdamUMCdb, a recently published large European ICU database, publishing open-access code for wider use by critical care researchers.

**Subjects**  Human and Biomedical Sciences, Medical Informatics, Physiology

**Submitted:**   18 December 2021

* Corresponding author. E-mail: te269@cam.ac.uk

Preprint submitted at dx.doi.org/10.5281/zenodo.6335285

## STATEMENT OF NEED

Sepsis is defined as life-threatening organ dysfunction resulting from a dysregulated host response to infection [1] and is a primary cause of critical illness and mortality. Early identification and treatment, which may include complex organ-support on the intensive care unit (ICU), is crucial for survival [2]. Historically, it has been been difficult to quantify both its incidence and mortality rate within the ICU, as it is a heterogeneous syndrome characterised by a wide-ranging infectious agent, infection site, treatment history and host response. Multiple organ dysfunction in septic patients follows from post-infection dysregulation in the immunology, biochemistry and physiology of the patient, and this in turn leads to morbidity and mortality. The third consensus definition of sepsis, by the Sepsis-3 Task Force [1], recommended a revised definition better aligned to this concept, explicitly incorporating mortality in order to ameliorate previous limitations and to allow for greater consistency in operationalising the definition criteria across different centres [3]. The Sepsis-3 clinical criteria are an acute increase in Sequential Organ Failure Assessment (SOFA) [4] of at least 2 points, accompanying a suspected or documented infection, with the criteria for septic shock further requiring both use of vasopressors and a lactate level >2 mmol/L.

## Sepsis-3

The third consensus definition of sepsis, by the Sepsis-3 Task Force [1], recommended a revised definition to address and ameliorate previous limitations and to allow for greater consistency in operationalising the definition criteria across different centres [3]. Nevertheless, there is not unanimous agreement within the intensive care community on the utility of the new definition [5, 6]. The Sepsis-3 clinical criteria are an acute increase in Sequential Organ Failure Assessment (SOFA) [4] of at least 2 points, accompanying a suspected or documented infection, with the criteria for septic shock further requiring both use of vasopressors and a lactate level >2 mmol/L. SOFA measures the severity of organ dysfunction across the six domains of the respiratory, neurological, cardiovascular, liver, coagulation and renal systems.

## Amsterdam UMC database

The most seriously ill patients with sepsis are treated on ICUs, which are perhaps the most data-dense clinical environments. Continuous multimodality monitoring, together with clinical expertise, forms the bedrock of patient care. However, the need to carefully balance the potential research benefits against patient privacy, ethics and legal concerns has limited the number of openly-accessible, de-identified large-scale databases of critical care patients to a small handful, such as Medical Information Mart for Intensive Care (MIMIC) [7] and eICU [8]. Differences in ICU demographics, resources, admission criteria and treatment strategies across different countries restrict the ability to generalise knowledge from these databases to other ICU populations and so comparative studies are crucial. Robust and well-documented cohort definition (sepsis, in this case considered here) is crucial to such reproducible large-scale observational data research. However a lack of uniform standard in data collection across different vendors and implementations of electronic health records hamper easy re-usability of models and code which is therefore necessarily database-specific.

Amsterdam University Medical Centers Database (AmsterdamUMCdb) [9] is a new, freely-accessible European ICU database, released in collaboration with the Society of Critical Care Medicine (SCCM) and the European Society of Intensive Care Medicine (ESICM). Compliant with both the U.S. Health Insurance Portability and Accountability Act (HIPAA) [10] and the European General Data Protection Regulation (GDPR) [11] through iterative risk-based patient de-identification, this database contains close to 1 billion data points from 20,109 critically ill patients admitted to Amsterdam UMC between 2003 and 2016. The database consists of patients admitted both to ICU and to the 'medium care unit' (MCU) in Amsterdam UMC. This data, comprised of seven comma-separated value tables, is combined from multiple systems in a 'data lake' structure linked through anonymised identifiers. AmsterdamUMCdb has already been the focus of several multidisciplinary research events, including two ESICM datathons [12] and a Neural Information Processing Systems (NeurIPS) privacy challenge [13].

## IMPLEMENTATION

## Sepsis-3 in Amsterdam UMC database

We provide a single script that computes the following: daily SOFA scores (individual components and total score) for each admission, antibiotic escalation on a daily basis, and finally sepsis/septic shock episodes (where one 'day' corresponds to each 24 h period after

admission). Our definition of each SOFA component score follows the AmsterdamUMCdb SOFA script, and we extend this computation from just the 24 h period post admission to a longer time period spanning multiple 'days'. This time period may be specified early within our script. Where no SOFA scores were available prior to ICU admission, the SOFA components were assumed to be zero, as per the Sepsis-3 recommendation. However, at least three missing SOFA components resulted in discarding any identification of sepsis or not for that admission/day.

Following [3], we define infection by an increase in the maximum rank of any antibiotics administered (or the number of antibiotics of maximum rank), according to the classification proposed by [14], where at least one antibiotic was given intravenously. This operationalisation of Sepsis-3 with a suspected infection identified only by antibiotic escalation is reliant on clinical judgement rather than a confirmed infection, which is a limitation of this approach. In our implementation, we have explicitly identified and disregarded routine or prophylactic administration of certain antibiotics within the standard procedure in Amsterdam UMC. For example, Amsterdam UMC practises selective digestive decontamination, which involves administering cefotaxime on admission (16 doses over four days) to everyone expected to stay at least one or two days. In Amsterdam UMC, cefotaxime is exchanged for ceftriaxone upon a suspected infection within this time period, so we have disregarded cefotaxime from the antibiotic escalation within the first four days.

A sepsis episode within a 24 h period was then defined as an increase in total SOFA score of at least two points between the previous and current, previous and subsequent, or current and subsequent 24 h periods, alongside an antibiotic escalation within that 24 h period. Finally, antibiotic use that accompanied admission after elective surgery were assumed to be prophylactic and as such was not classified as sepsis either on that 24 h period or the subsequent 24 h period. Any subsequent 24 h period that met the Sepsis-3 definition was however identified as a sepsis episode. Septic shock was defined as a subset of sepsis episodes with a cardiovascular SOFA score of at least 3 (i.e. using vasopressors) and a maximum lactate of at least 2 mmol/L. We assumed that vasopressors were administered if required to maintain a mean arterial pressure at least 65 mmHg, assuming adequate fluid administration.

The accompanying AmsterdamUMCdb GitHub repository [15] contains descriptions of the data structure and instructions for querying the database, as well as Python scripts for extracting and checking crucial concepts, such as primary admission diagnosis and severity scores within the first 24 h. Noting that sepsis at admission is rarely documented consistently, the definition of sepsis in the AmsterdamUMCdb scripts is given by one of the following criteria:

- sepsis at admission flagged in the admission form by the attending clinician
- the admission diagnosis, medical or surgical, is considered a severe infection, e.g., gastrointestinal perforation, cholangitis, meningitis
- non-prophylactic use of antibiotics after surgery
- use of antibiotics and cultures drawn within 6 h of admission.

These criteria are generally less consistent than Sepsis-3. Of 20,091 unique first admissions to ICU, in which a diagnosis of sepsis within the 24 h period before or the 24 h



**Table 1.** Confusion matrix for the current AmsterdamUMCdb sepsis definition, compared to Sepsis-3.

| Unique first admissions | | Current | | All admissions | | Current | |
|---|---|---|---|---|---|---|---|
| | | **True** | **False** | | | **True** | **False** |
| Sepsis-3 | True | 2114 | 4319 | Sepsis-3 | True | 2410 | 5145 |
| | False | 838 | 12,820 | | False | 996 | 14,533 |

There are 25 admissions in total that have insufficient data for a Sepsis-3 diagnosis.
Sensitivity for first admissions only is 0.33 and specificity is 0.94. Sensitivity for all admissions is 0.32 and specificity is 0.94.

**Table 2.** Example SOFA score output, containing the SOFA component and total scores.

| admissionid | time | sofa_respiration_score | sofa_coagulation_score | sofa_liver_score | sofa_cardiovascular_score | sofa_cns_score | sofa_renal_score | sofa_total_score |
|---|---|---|---|---|---|---|---|---|
| 0 | −1 | NaN | 0 | NaN | NaN | NaN | 0 | 0 |
| 0 | 0 | 3 | 0 | NaN | 1 | 0 | 0 | 4 |
| 0 | 1 | 2 | 1 | NaN | 2 | NaN | 0 | 5 |
| 1 | −1 | NaN | 1 | NaN | NaN | NaN | 0 | 1 |
| 1 | 0 | 2 | 0 | NaN | 2 | 0 | 0 | 4 |
| 1 | 1 | NaN | NaN | NaN | 0 | NaN | 0 | 0 |
| 2 | −2 | NaN | 0 | 0 | NaN | NaN | 0 | 0 |
| 2 | −1 | NaN | 1 | NaN | NaN | NaN | NaN | 1 |
| 2 | 0 | 2 | 0 | NaN | 4 | 0 | 0 | 6 |
| 3 | −3 | NaN | 0 | NaN | NaN | NaN | 1 | 1 |
| 3 | 0 | 2 | 0 | NaN | 0 | NaN | 1 | 3 |

The column 'time' denotes the 'day' of admission, which is the 24 h period after the ICU/MCU admission. A negative 'time' indicates data prior to ICU admission (i.e. a partial SOFA score from when the patient was in a general ward prior to transfer to ICU).
NaN indicates that this SOFA component was not measured or could be calculated from the data available.
The total SOFA score is the sum of the components, with NaN values replaced by 0, as per [1, 4].

period after admission could be made via the Sepsis-3 definition, the sensitivity of the above current criteria compared to the Sepsis-3 in the first 24 h is poor (Table 1). Furthermore, the previous script is designed with admission in mind only, and does not identify sepsis episodes or septic shock outside of the first 24 h period.

To keep our code agnostic of choice of database management system, we work directly with the underlying tables in comma-separated value (CSV) format. The output of this script are two additional CSV files, of a similar size to the base 'admissions' table (<10 MB), one containing all SOFA scores for each admission/day and the other containing binary indicators of the following for each admission/day: total SOFA score, antibiotic escalation, prophylaxis, infection, sepsis episodes and septic shock. These output tables are described in Tables 2 and 3, and further details about the implementation are documented within the code.

## AVAILABILITY OF SOURCE CODE AND REQUIREMENTS

- Project name: Sepsis-3 in AmsterdamUMCdb
- Project home page: https://github.com/tedinburgh/sepsis3-amsterdamumcdb
- Operating system(s): Platform independent
- Programming language: Python 3.7.9
- Other requirements: Python modules – numpy 1.20.3 or higher, pandas 1.2.5 or higher, re 2.2.1 or higher, amsterdamumcdb (installation via/described in [15])
- License: MIT License
- RRID:SCR_022042.

**Table 3.** Example sepsis table output, containing also the total SOFA score, infection status, prophylactic use of antibiotics and septic shock.

| admissionid | time | sofa_total_score | antibiotic_escalation | prophylaxis | infection | sepsis_episode | septic_shock |
|---|---|---|---|---|---|---|---|
| 0 | −1 | 0 | True | True | False | False | False |
| 0 | 0 | 4 | False | False | False | False | False |
| 0 | 1 | 5 | NaN | False | False | False | False |
| 1 | −1 | 1 | True | True | False | False | False |
| 1 | 0 | 4 | False | False | False | False | False |
| 1 | 1 | 0 | NaN | False | False | False | False |
| 2 | −2 | 0 | NaN | False | False | False | False |
| 2 | −1 | 1 | True | False | True | True | False |
| 2 | 0 | 6 | False | False | False | False | False |
| 3 | −3 | 1 | NaN | False | False | False | False |
| 3 | −1 | 0 | True | False | True | True | False |
| 3 | 0 | 3 | False | False | False | False | False |

The column 'time' denotes the 'day' of admission, which is the 24 h period after the ICU/MCU admission.
NaN in the column 'antibiotic_escalation' indicates that this 24 h period occurs before any antibiotics were first administered.
Antibiotic escalation in elective post-operative admissions was assumed to be a prophylactic increase in antibiotic administration, rather than an antibiotic escalation due to an infection. Post-operative s are also likely to have a high SOFA score because of surgery.
A sepsis episode is defined as antibiotic escalation accompanied by an increase in SOFA score of 2 or more. This increase in SOFA can either be over the previous and current 'days', the current and subsequent 'days', or the previous and subsequent 'days'. For the sepsis episode in this table (admissionid 2, time −1), the increase in SOFA from 1 to 6 from day −1 to day 0 is accompanied by an antibiotic escalation on day −1.

## DATA AVAILABILITY

The dataset supporting the results of this article (AmsterdamUMCdb) is freely-accessible. Although de-identified, it still contains detailed information regarding the clinical care of patients, so must be treated with appropriate care and respect and cannot be shared without going through a formal application process. Access to the database is on request through [16], under moderate conditions, including completion of a standard training course for handling de-identified clinical data. Snapshots of the GitHub repositories and forms and documentation for applying for access to the data are available via the *GigaScience* GigaDB repository [17]. To gain access to AmsterdamUMCdb requires the following steps:

(1)  Users must first complete the Data or Specimens Only Research (DSOR) course from CITI (https://about.citiprogram.org/).
(2)  Users must then submit a signed copy of the AmsterdamUMCdb application form (see forms in GigaDB [17]).
(3)  Once the application form has been approved, users must create an account on EASY (https://easy.dans.knaw.nl/ui/home), complete their user profile and request download permission for the dataset on EASY.
(4)  Once registered, users should then contact DANS (Data Archiving and Networked Services), who should send a link to the AmsterdamUMCdb archive.

## DECLARATIONS
## LIST OF ABBREVIATIONS

Amsterdam University Medical Centers database: AmsterdamUMCdb; comma-separated value: CSV; European Society of Intensive Care Medicine: ESICM; General Data Protection Regulation: GDPR; Health Insurance Portability and Accountability Act: HIPAA; intensive care unit: ICU; Medical Information Mart for Intensive Care: MIMIC; medium care unit: MCU; Sequential Organ Failure Assessment: Neural Information Processing Systems: NeurIPS; SOFA; Society of Critical Care Medicine: SCCM; Structured Query Language: SQL.

## ETHICAL APPROVAL

Ethical approval for the data collection, deidentification and governance are described in [9]. No additional ethical approval was required for this manuscript.

## COMPETING INTERESTS

The authors declare that they have no competing interests.

## FUNDING

TE is funded by Engineering and Physical Sciences Research Council (EPSRC) National Productivity Investment Fund (NPIF) EP/S515334/1, reference 2089662. The funding body had no input in the study design, data collection, analysis or manuscript.

## AUTHOR'S CONTRIBUTIONS

TE, SE and AE contributed conceptualisation. PE and PT were responsible for data curation and project administration. TE and PT developed software. SE and AE provided supervision. TE and AE drafted the manuscript. All authors read and approved the final version of the manuscript.

## ACKNOWLEDGEMENTS

We would like to thank the reviewers, Chris Armit and Tom Pollard, for their useful comments on the manuscript and about the code in the open peer review for GigaByte.

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
