## [Reviewer Report]

Comments on revised manuscriptThis Technical Release manuscript describes Sepsis-3 criteria in AmsterdamUMCdb. I thank the authors for addressing my comments and for updating the linked GitHub repository. The authors have explained that streamlining access to AmsterdamUMCdb is unfortunately outwith their control, hence the need for a user to gain access via EASY registration. This is completely understandable and I thank the authors for their explanation.

I recommend this Technical Release manuscript for publication in GigaByte.

---

## [Reviewer Report]

Reviewer name and names of any other individual's who aided in reviewerChris ArmitDo you understand and agree to our policy of having open and named reviews, and having your review included with the published manuscript. (If no, please inform the editor that you cannot review this manuscript.)YesIs the language of sufficient quality?YesPlease add additional comments on language quality to clarify if neededA minor point is that the contents of the Restricted Access AmsterdamUMCdb tabular data files are in Dutch, and so researchers may have to familiarise themselves with some of the terms that are used in the data files e.g. Neurochirurgie = Neurosurgery, Vaatchirurgie = Vascular Surgery, Verloskunde = Midwifery etc.Is there a clear statement of need explaining what problems the software is designed to solve and who the target audience is? YesAdditional CommentsThis Technical Release manuscript describes Sepsis-3 criteria in AmsterdamUMCdb. Sepsis-3 is the third consensus definition of sepsis, which includes a revised definition of sepsis based on a scoring system that utilises Sequential Organ Failure Assessment (SOFA). SOFA measures the severity of organ dysfunction across six domains including respiratory, neurological, cardiovascular, liver, coagulation and renal systems. AmsterdamUMCdb is a freely accessible intensive care database with de-identified health data relating to intensive care unit admissions, and includes demographics, vital signs, laboratory tests and medications. Is the source code available, and has an appropriate Open Source Initiative license <a href="https://opensource.org/licenses" target="_blank">(https://opensource.org/licenses)</a> been assigned to the code?YesAdditional CommentsThe manuscript is well written, and the software is publicly available from GitHub (https://github.com/tedinburgh/sepsis3-amsterdamumcdb), where it has been ascribed an OSI-approved MIT license.As Open Source Software are there guidelines on how to contribute, report issues or seek support on the code?YesAdditional CommentsThe GitHub archive was created by Dr Tom Edinburgh, who is the contact author for this manuscript.Is the code executable?YesAdditional CommentsIs installation/deployment sufficiently outlined in the paper and documentation, and does it proceed as outlined?YesAdditional CommentsIs the documentation provided clear and user friendly?YesAdditional CommentsThe AmsterdamUMCdb Test Data is Restricted Access. I outline below the various steps that were needed to access the Test Data.

• To gain access to AmsterdamUMCdb, one must first complete the Data or Specimens Only Research (DSOR) course from CITI
• Reviewers must then submit a signed copy of application form arfeula_v1.6.pdf
• This application form and the link to the DSOR course are available from the following link:
• https://www.amsterdammedicaldatascience.nl/#amsterdamumcdb

• This application must be counter-signed by an intensivist i.e. a health professional who specialises in the care of critically ill patients, and who is the named reference on the application from

Once the application form has been approved, reviewers must do the following:

• Create an account on EASY using the same institutional e-mail address as on their form
• Complete their user profile in EASY, including organisation and department
• Request download permission for the dataset on EASY

Once registered with EASY, reviewers will be granted access to the AmsterdamUMCdb Data File that is archived in EASY: 

• https://easy.dans.knaw.nl/ui/datasets/id/easy-dataset:130980 

There is currently a single file archived on this link. This single file is a PDF file that states the following:

• Your request has been granted, the dataset has been archived outside EASY due to the size.
• Please contact DANS at info@dans.knaw.nl for delivery of the dataset to you. Preferably mention the citation of the dataset:
• Elbers, Dr. P.W.G. (Amsterdam UMC) (2019): AmsterdamUMCdb v1.0.2. DANS. https://doi.org/10.17026/dans-22u-f8vd

Reviewers should then contact DANS (Data Archiving and Networked Services), who should send a SURFfilesender link to the 8.3GB AmsterdamUMCdb-v1.0.2.zip archive. This link is active for 3 weeks.

The zip file contains seven tabular data files (CSV file format) that I list below.

• admissions.csv
• drugitems.csv
• freetextitems.csv
• listitems.csv
• numericitems.csv
• procedureorderitems.csv
• processitems.csv
Is there a clearly-stated list of dependencies, and is the core functionality of the software documented to a satisfactory level?YesAdditional CommentsHave any claims of performance been sufficiently tested and compared to other commonly-used packages? Not applicableAdditional CommentsAre there (ideally real world) examples demonstrating use of the software? YesAdditional CommentsIs automated testing used or are there manual steps described so that the functionality of the software can be verified?YesAdditional CommentsAny Additional Overall Comments to the AuthorThis Technical Release manuscript describes Sepsis-3 criteria in AmsterdamUMCdb. Sepsis-3 is the third consensus definition of sepsis, which includes a revised definition of sepsis based on a scoring system that utilises Sequential Organ Failure Assessment (SOFA). SOFA measures the severity of organ dysfunction across six domains including respiratory, neurological, cardiovascular, liver, coagulation and renal systems. AmsterdamUMCdb is a freely accessible intensive care database with de-identified health data relating to intensive care unit admissions, and includes demographics, vital signs, laboratory tests and medications. The manuscript is well written, and the software is publicly available from GitHub (https://github.com/tedinburgh/sepsis3-amsterdamumcdb), where it has been ascribed an OSI-approved MIT license.

The AmsterdamUMCdb Test Data is Restricted Access. I am grateful to the authors for providing me with access to this very significant data resource. I outline below the various steps that were needed to access the Test Data.

• To gain access to AmsterdamUMCdb, one must first complete the Data or Specimens Only Research (DSOR) course from CITI
• Reviewers must then submit a signed copy of application form arfeula_v1.6.pdf
• This application form and the link to the DSOR course are available from the following link:
• https://www.amsterdammedicaldatascience.nl/#amsterdamumcdb

• This application must be counter-signed by an intensivist i.e. a health professional who specialises in the care of critically ill patients, and who is the named reference on the application from

Once the application form has been approved, reviewers must do the following:

• Create an account on EASY using the same institutional e-mail address as on their form
• Complete their user profile in EASY, including organisation and department
• Request download permission for the dataset on EASY

Once registered with EASY, reviewers will be granted access to the AmsterdamUMCdb Data File that is archived in EASY: 

• https://easy.dans.knaw.nl/ui/datasets/id/easy-dataset:130980 

There is currently a single file archived on this link. This single file is a PDF file that states the following:

• Your request has been granted, the dataset has been archived outside EASY due to the size.
• Please contact DANS at info@dans.knaw.nl for delivery of the dataset to you. Preferably mention the citation of the dataset:
• Elbers, Dr. P.W.G. (Amsterdam UMC) (2019): AmsterdamUMCdb v1.0.2. DANS. https://doi.org/10.17026/dans-22u-f8vd

Reviewers should then contact DANS (Data Archiving and Networked Services), who should send a SURFfilesender link to the 8.3GB AmsterdamUMCdb-v1.0.2.zip archive. This link is active for 3 weeks.

The zip file contains seven tabular data files (CSV file format) that I list below.

• admissions.csv
• drugitems.csv
• freetextitems.csv
• listitems.csv
• numericitems.csv
• procedureorderitems.csv
• processitems.csv

A minor point is that the contents of the tabular data files are in Dutch, and so researchers may have to familiarise themselves with some of the terms that are used in the data files e.g. Neurochirurgie = Neurosurgery, Vaatchirurgie = Vascular Surgery, Verloskunde = Midwifery etc.

I was wondering whether it would it be possible to streamline the application process for AmsterdamUMCdb? For example, as the AmsterdamUMCdb dataset is archived outside of EASY, is it necessary to register with EASY? It may be swifter for a researcher if this step was omitted and they were invited to contact DANS at an earlier step in the process.
RecommendationAccept

---

## [Reviewer Report]

Reviewer name and names of any other individual's who aided in reviewerTom PollardDo you understand and agree to our policy of having open and named reviews, and having your review included with the published manuscript. (If no, please inform the editor that you cannot review this manuscript.)YesIs the language of sufficient quality?YesPlease add additional comments on language quality to clarify if neededOverall the paper is clear and well written.Is there a clear statement of need explaining what problems the software is designed to solve and who the target audience is? YesAdditional CommentsIs the source code available, and has an appropriate Open Source Initiative license <a href="https://opensource.org/licenses" target="_blank">(https://opensource.org/licenses)</a> been assigned to the code?YesAdditional Comments- The code is currently in a personal GitHub repo. The authors could consider using a service like Zenodo to assign a persistent URL to the code, which could be used in place of the github URL. This may help to ensure the code is persistently available (e.g. in the case that the author decides to move the code from GitHub to a new location).As Open Source Software are there guidelines on how to contribute, report issues or seek support on the code?NoAdditional Comments- It would be helpful for the paper to describe how the user community will be supported. How would the authors like community members to report bugs, fix bugs, contribute improvements, etc? 
- The authors could briefly discuss this in the paper, and perhaps add guidelines to the repository (e.g. in a CONTRIBUTING.md file: https://github.com/github/docs/blob/main/CONTRIBUTING.md).Is the code executable?YesAdditional Comments- It would be great if example or synthetic version of the AmsterdamUMCdb was provided in the repository. At its simplest, this could be comprise of empty tables. This would allow users to easily test the code against the sample data. It would also support development of a testing framework.Is installation/deployment sufficiently outlined in the paper and documentation, and does it proceed as outlined?YesAdditional CommentsIs the documentation provided clear and user friendly?NoAdditional CommentsThe paper is well written. The code itself would benefit from significant work to improve readability and usability (for example, with the use of functions and docstrings). I have outlined some suggestions below:
- The following link gives a nice example of a well formatted script: https://www.annasyme.com/docs/python_structure.html. There are many reasons why this structure is helpful (e.g. readability; facilitates import and reuse of code; avoids hardcoding of arguments; etc)
- Following standard style guidelines improves readability makes it easier to identify bugs (e.g. PEP8: https://www.python.org/dev/peps/pep-0008/). Installing a formatter allows the style to be consistently applied across the code.
- Refactoring the code to use functions, classes, and modules, as appropriate would help readability and enable unit testing. https://github.com/tedinburgh/sepsis3-amsterdamumcdb/blob/main/concepts/sepsis3/reason_for_admission.py is >1300 lines with a single function.
- Instead of hardcoded variables like "../../data/additional_files/", these could be handled as arguments with defaults (e.g. with argparse). Is there a clearly-stated list of dependencies, and is the core functionality of the software documented to a satisfactory level?NoAdditional Comments- The dependencies are fairly clear, but the code would benefit from refactoring for readability as dicussed.
- Presumably the database has some kind of version control? Is there a plan to keep the code in sync with the database versions? Have any claims of performance been sufficiently tested and compared to other commonly-used packages? NoAdditional CommentsThe paper could be improved by providing a more in depth analysis of the sepsis cohort that can extracted using this code. e.g. how do characteristics of patients with sepsis-3 differ from those without? Although not necessary, these characteristics could be compared to similar cohorts extracted from the other datasets mentioned in the paper.Are there (ideally real world) examples demonstrating use of the software? YesAdditional CommentsThe benefits of making this code available to the research community are clear.Is automated testing used or are there manual steps described so that the functionality of the software can be verified?NoAdditional Comments- The code does not include a testing framework, and the current implementation is not well suited to testing
- If refactoring the code as discussed above, then the authors could consider adding some simple unit tests at the same time.Any Additional Overall Comments to the Author# Summary
The paper describes implementation of open source python scripts for computing sepsis-3 for patients in AmsterdamUMCdb, a publicly accessible critical care database. The paper is well written and the efforts are well motivated. The code would benefit from a significant clean up.

# Strengths
- Many analyses that use AmsterdamUMCdb are likely to benefit from open source code for computation of Sepsis-3.
- The paper is well written and demonstrates that significant thought has gone into the implementation, taking into account both clinical and technical perspectives.
- The code is shared with an appropriate license, in a public, version controlled repository.

# Suggestions for improvement
- The code would benefit from a fairly significant clean up. Currently, in my opinion it does not follow standard practice for Python and as a result it is tricky to read, reuse, and iteratively improve.RecommendationMajor Revisions